**Data Availability Statement:** Data are from the Swiss Mongolian pediatric Project whose authors

# Comparison of quality and interpretation of newborn ultrasound screening examinations for developmental dysplasia of the hip by basically trained nurses and junior physicians with no previous ultrasound experience

Munkhtulga Ulziibat[1,2]*, Bayalag Munkhuu[2], Raoul Schmid[3], Corinne Wyder[4], Thomas Baumann[5], Stefan Essig[5]

1 Department of Health Sciences and Medicine, University of Lucerne, Lucerne, Switzerland, 2 National Center for Maternal and Child Health, Ulaanbaatar, Mongolia, 3 Baarer Kinderarzt Praxis, Baar, Switzerland, 4 Kinderärzte KurWerk, Burgdorf, Switzerland, 5 Center of Primary and Community Care, Department of Health Sciences and Medicine, University of Lucerne, Lucerne, Switzerland

* umunhtulga@gmail.com

## Abstract

### Background

We are obliged to give babies the chance to profit from a nationwide screening of developmental dysplasia of the hip in very rural areas of Mongolia, where trained physicians are scarce. This study aimed to compare the quality and interpretation of hip ultrasound screening examinations performed by nurses and junior physicians.

### Methods

A group of 6 nurses and 6 junior physician volunteers with no previous ultrasound experience underwent Graf's standard training in hands-on practice. Newborns were examined before discharge from the hospital, according to the national guideline. Two standard documentation images of each hip were saved digitally. The groups were compared on the proportion of good quality of sonograms and correct interpretation. Two Swiss supervisors' agreed diagnosis according to Graf was considered the final reference for the study purposes.

### Results

A total of 201 newborns (402 hips or 804 sonograms) were examined in the study, with a mean age of 1.3±0.8 days at examination. Junior physicians examined 100 newborns (200 hips or 400 sonograms), while nurses examined 101 newborns (202 hips or 404 sonograms). The study subjects of the two groups were well balanced for the distribution of baseline characteristics. The study observed no statistically significant difference in the quality of Graf's standard plane images between the providers. Eventually, 92.0% (92) of the physician group and 89.1% (90) of the nurse group were correctly diagnosed as "Group A" (Graf's

may be contacted via its website: www.smopp.ch. The authors will submit their anonymized all data file in stata format (.dta).

**Funding:** The first author received doctoral training support from the Swiss Mongolian Pediatric Project (SMOPP) http://www.sipp.swiss/ and the Swiss Association of Pediatric Ultrasound (SVUPP). Moreover, ultrasound machines and examination cradles were provided by SMOPP free of charge under the governmental national screening program. The funders had no role in study design, data collection and analysis, decision to publish, or preparation of the manuscript.

**Competing interests:** The authors have declared that no competing interests exist.

Type 1 hip) or "Non-Group A" hips (p = 0.484). The most common errors among the groups were a missing lower limb, wrong measurement lines, and technical problems.

## Conclusion

Our study provides evidence that while there might be a trend of slightly more technical mistakes in the nurse group, the overall diagnosis accuracy is similar to junior physicians after receiving standard training in Graf's hip ultrasound method. However, after basic training, regular quality control is a must and all participants should receive refresher trainings. More specifically, nurses need training in the identification of anatomical structures.

## Introduction

Developmental dysplasia of the hip (DDH) is one of the most common disorders of the osteoarticular system with public health priorities in otherwise healthy children. The reported incidence of DDH are quite variable depending on detection methods, ages and diagnostic criteria [1, 2]. Mongolian studies reported a 1–2% incidence of DDH among neonates by ultrasound with Graf's technique [3], an incidence comparable to that in European neonates [4, 5]. DDH is a multifactorial disorder. Contributing factors for the development of DDH are genetic and non-genetic factors [6, 7]. Early diagnosis of DDH in neonates is the most critical obligation for achieving the best outcome, using a simple, non-surgical method in the shortest possible time [8–11]. Delayed diagnosis results in losing the potential to remodel the acetabular roof, lengthening the treatment duration, and may cause lifelong disability [12, 13].

The hip ultrasound, according to Graf, which provides basic information about the biomechanical situation, is widely used in several countries as a gold standard of the primary tool in screening for DDH. Unlike other methods, it is standardized, easy to perform, and reproducible. The non-surgical treatment is usually based on the degree of severity of DDH [3, 14]. Graf's method has been shown to be sensitive and specific for the early diagnosis of DDH [15]. Studies have also shown that ultrasonographic screening by Graf is cost-effective [16]. The method uses a coronal ultrasound image through the center of the acetabulum (standard plane) and the hip is evaluated by measuring two angles formed by three lines drawn from three landmarks (lower limb or the bottom of the acetabulum, the acetabular labrum and the lateral edge of the acetabulum). Subsequently, the bony roof angle (alpha angle) and the cartilage roof angle (beta angle) are measured and the hip joint is classified (**Fig 1**).

The introduction of Graf's method of hip ultrasonography and more effective non-surgical treatment in Mongolia since 2010 changed the policy and time of diagnosis and preventive treatment in children with DDH, resulting in more prevented cases of childhood disabilities [4, 5]. In 2017, the Mongolian Government approved the hip ultrasound screening as a nationwide screening program. This screening program aims to screen every newborn baby using hip ultrasound by Graf's method and to provide early prevention treatment to eliminate childhood disabilities due to DDH in the country. All major 7 maternity hospitals across Ulaanbaatar, the capital city, and all 21 provincial (aimag) general hospitals are equipped with the necessary equipment and materials; and doctors trained in the method by the Swiss-Mongolian Pediatric Project (SMOPP), a humanitarian aid project. Used Tübingen braces are collected in Switzerland and German-speaking countries and are being reused as a way to help correct the skeletal system.

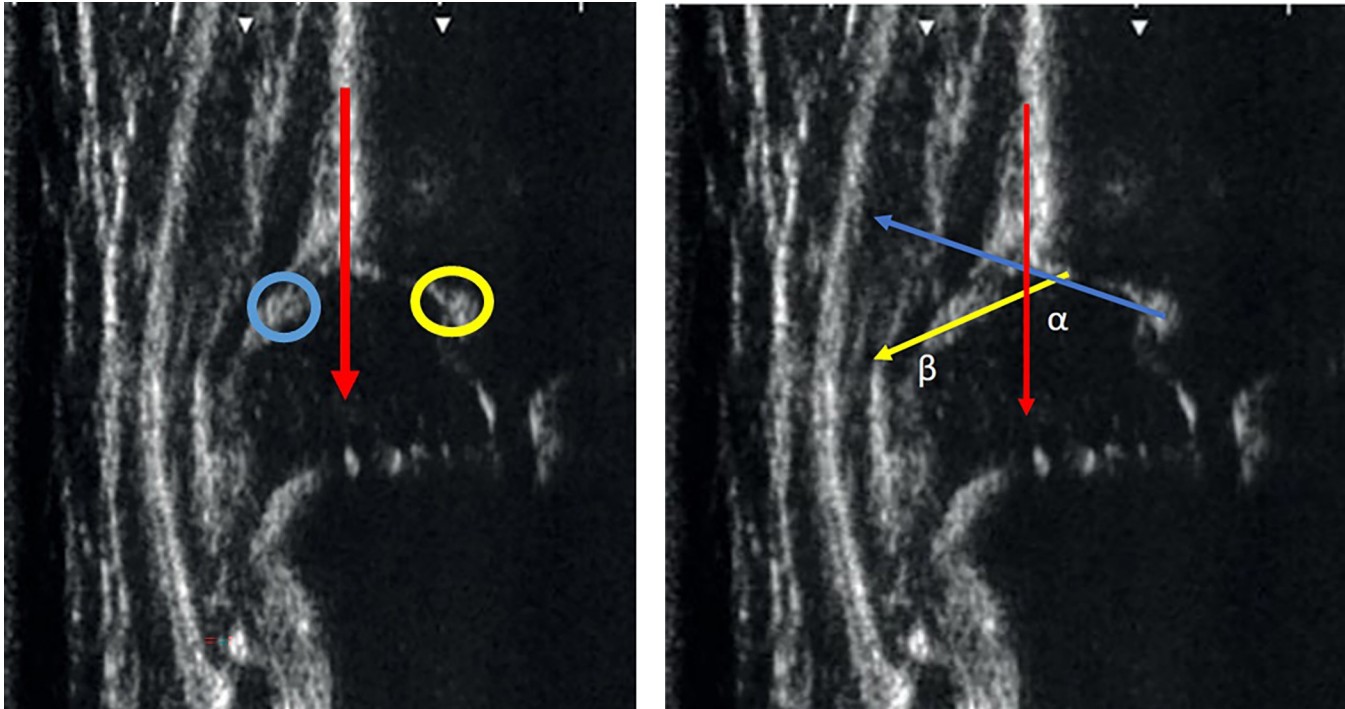

**Fig 1. A hip sonogram.** A. Three landmarks of the standard plane. 1. Standard cut (red arrow), 2. Lower limb or the bottom of the acetabulum (yellow circle), 3. Acetabular labrum and the lateral edge of the acetabulum (blue circle). B. Hip is evaluated by measuring two angles formed by three lines drawn from three landmarks. 1. Basic line (red), 2. Bony roof line (blue), 3. Cartilaginous roof line (yellow).

In Mongolia, a web-based, password-protected platform is used for quality control purpose. The tool enables screeners in all hospitals to upload digital images and annotations exported from the ultrasound machines. Four images (two per hip side; one of them with measured alpha and beta angles according to Graf), a unique identification number, age, sex and diagnosis are required for all examinations. After the upload, quality is controlled by a local expert (a doctor with good knowledge of the Graf method and at least three years of experience of hip ultrasound and treatment at a provincial hospital). This allowes continuous and reliable reviews of all examinations. In addition, Mongolian supervisors (doctors with profound knowledge of the Graf method and more than 10 years of experience of hip ultrasound and treatment in UB) check examinations from all hospitals and promptly send comments to the experts and screeners.

Since 2019, the screening project has been expanded to major sub-provincial (soum) hospitals responsible for deliveries as a primary source of medical care for nomadic rural residents. However, the sub-provincial hospitals face many significant and unique barriers when considering the widespread adoption of newer innovations in health care, such as hip ultrasound screening in newborns. One barrier is a lack of trained physicians capable of performing hip ultrasounds (huge turnover within the doctors' teams, loss of trained doctors due to high internal and international migrations etc.) like in other developing countries [17]. Therefore, we felt obliged to give the babies in very rural areas the chance to profit from the nationwide screening program. We searched for other possibilities to examine all newborn babies of nomadic families. This study was performed after 3 years of piloting the expansion to rural sub-provincial areas. We aimed to compare the quality and interpretation of newborn hip ultrasound examinations (Graf's Type 1 versus "Non-type 1"hips) between basically trained

nurses with no previous ultrasound experiences and trained junior physicians with no previous ultrasound experience.

## Materials and methods

### Study design and settings

A prospective cross-sectional study was conducted in 3 hospitals: a referral hospital in Ulaanbaatar, a provincial (aimag) general hospital, and a sub-provincial (soum) hospital from March to December 2021. The Ethics Review Committee of the Ministry of Health of Mongolia (MOH), reviewed and approved the study protocol.

### Study participants

A sample of 6 nurses and 6 junior physicians were nurse and junior physician volunteers in the selected hospitals. The only selection criterion was that they had no previous knowledge or experience in ultrasound. The six nurses were all certified with bachelor degree (4 years of undergraduate training) from Nursing school of Mongolian National University of Medical Science and had 3–4 years (4 nurses- 3 years and 2 nurses- 4 years) of experience. The six junior physicians were first-year pediatric resident doctors and all received undergraduate training in medicine (six years) from Mongolian National University of Medical Science. All junior physicians had 1–2 years of clinical experience in primary care.

All consecutive newborns born at the 3 hospitals during the study period were ultrasonographically screened by six nurses (two from each hospital) and six junior physicians (two from each hospital). Exclusion criteria were severe or very severe conditions, congenital anomalies, and refusal of parents through a written informed consent.

### Preparatory phase

Nurse- and junior physician- volunteers who consented to participate in the study were trained on using the ultrasound machine through hands-on instruction. Two supervisors with extensive experience in hip sonography and the "ABCD" system facilitated the training. The "ABCD" system is a diagnostic and therapeutic framework based on Graf's technique [18]. SMOPP established the system as a nationally recognized standard, ensuring it is user-friendly for screeners in maternity hospitals and capable of distinguishing between groups essential for both controls and treatment. The system is entirely based on the method described by Graf and translates his initial differentiation into four groups: Group A includes Graf Type 1; Group B includes Graf Type 2a; Group C includes Graf Types 2c, D and 3; and Group D includes Graf Type 4 (**Table 1**).

The nurses and junior physicians received training in basic hip sonography, with emphasis on ultrasound principles, macro, and sono-anatomy of the hip joint, obtaining standard images, and "ABCD" system, measurements, tilting errors (inaccuracies due to improper transducer positioning resulting in distortions and misleading images), pitfalls and counseling of parents. The standard week courses (15 hours including practical works) were organized in March in sub-provincial, August in provincial, and September 2021 in Ulaanbaatar hospitals. There was no pre-training test since all participants had no previous knowledge or experience in Graf's method of hip ultrasound examinations. However, a post-training test was performed and included questions on theoretical and practical (interpretation and handling) knowledge.

Following the standard introductory course, the nurses and junior physicians completed a week of hip sonography screening and SMOPP's "HipScreen" quality control online platform practice in their hospitals, supervised by the supervisors who provided the training. Each

Table 1. Comparison of Graf types and ABCD groups for hip ultrasound.

| Graf type | Graf angle $\alpha$ (°) | Graf angle $\beta$ (°) | Graf therapy | ABCD group | ABCD angle $\alpha$ (°) | ABCD therapy |
|---|---|---|---|---|---|---|
| 1a | > 60 | < 55 | None | A | > 60 | None |
| 1b | | > 55 | None | | | |
| 2a | 50–59 | | Control | B | > 50 ≤ 60* | Control |
| 2a+ | | | Control | | | |
| 2a- | | | Spreading device/Pavlik | C | < 50 | Tübingen braces |
| 2b | | | | | | |
| 2c stable | 43–49 | < 77 | | | | |
| 2c unstable | | | Plaster | | | |
| D | | > 77 | | | | |
| 3a | < 43 | | Extension | | | |
| 3b | | | | | | |
| 4 | | | Operation | D | Measurement is impossible | Tübingen braces/Operation |

* Group B hips were defined as angle alpha minus the age in weeks = between 50˚-60˚

nurse and junior physician undertook 10 newborns' hip sonograms (2 for each hip, totaling 80 sonograms), focusing on gaining experience and competency in identifying children who require a referral (non-Group A hips) before testing their skills.

## Data collection

Newborns were examined on a daily basis before discharge from the hospital (1st or 2nd days after birth) according to the national guideline [19]. Those who met the selection criteria were recruited in the study, and the following procedures were done:

1. counseling of mothers and brief information about DDH

2. hip ultrasound examination

3. information about the next steps according to hip ultrasound examination results.

All eligible newborns were examined using a GE LOGIQ series (C3) ultrasound machine with a 7 to 8-MHz linear-array transducer. The examinations were performed and interpreted according to the national guideline based on the SMOPP's "ABCD" system [18], the modified Graf's system by the trained nurses or junior physicians depending on the days (odd days for nurses and even days for the junior physicians). Two standard documentation images of each hip were saved digitally using the HipScreen system. Monthly ultrasound follow-up control until full maturation and management done by both groups, depending on the results of the ultrasounds, followed the national guideline [19].

Two Swiss supervisors checked all the nurses' and junior physicians' hip ultrasound images. The two supervisors were blinded to the nurses and doctors. All supervisors are internationally certified and have at least 10 years of experience in Graf's method of hip ultrasonography. Their final agreed diagnosis was considered the final reference for the study purposes. All supervisors interpreted all examinations (nurses and junior physicians); if discordant, a posterior discussion led to a consensual agreement.

## Outcome measurement

1. The following 4 indicators measured the quality of the newborn hip ultrasound images:

a.  Anatomical identification is "correct" if all anatomical structures, according to Graf [3, 10] are identified in the hip sonogram. If at least one structure is missing, the sonogram is considered "incorrect".

b.  The standard plane is "correct" if all 3 points according to Graf's criteria [3, 10] (Lower limb of the bony ilium, Mid-portion of the acetabular roof, and Acetabular labrum) are visible in the hip sonogram. If at least one structure is missing, the sonogram is considered "incorrect".

c.  Three lines (baseline, bony, and cartilaginous roof lines) are measured "correct" or "incorrect" according to Graf (baseline–a vertical line, parallel to the ossified lateral wall of the ilium; bony roof line–a line drawn from the inferior edge of the osseous acetabulum, the inferior iliac margin, at the triradiate cartilage roof to the most lateral point on the ilium, the superior osseous rim; and cartilaginous roof line–a line traced along the cartilaginous acetabulum's roof, extending from the acetabulum's lateral osseous edge to the center of the labrum) [3, 10, 20].

d.  The tilting was measured as "yes" or "no" depending on the absence/presence of any tilting errors according to Graf [3, 10]. In order to get a standard plane sonogram the transducer should be placed vertically on the hip joint. Tilted positions of the transducer can lead to significant examinations errors (usually a mechanical transducer guide prevents tilting errors by reducing the degrees of variability in positioning the transducer).

2.  The correct interpretation: Based on results of hip sonography the nurses and junior physicians were requested to arrive at a final diagnosis, utilizing one of the terms: "Group A" (Graf's Type 1 hip) or "Non-Group A" (Graf's non-Type 1 hip). Then it was checked against the reference interpretation performed by the Swiss experts. The interpretation was defined as "correct" or "incorrect".

## Statistical analyses

Statistical analyses were performed using Stata 16 (Stata Corporation) software. Continuous variables were summarized as mean, standard deviation, and non-continuous variables as frequencies and percentages. A comparison of two groups (nurses and junior physicians) was performed using the $\chi 2$ test, Fisher's exact test, and Student's t-tests depending on the variables, with a significance level probability of 0.05 or less.

Univariate comparison of correct and incorrect diagnosis was performed using Fisher's exact test. In the main analysis, we used the child and not the hip sonogram (2 sonograms per hip or 4 hip sonograms per child) as the unit of analysis. If a child had hip sonograms with different qualities, we evaluated the child based on the worst quality.

Power calculations indicate that we need to include at least 92 subjects in each group based on the assumptions that 1) the difference in quality of ultrasound is 15%, 2) the alpha level is 0.05, and 3) the required power is 80%.

## Results

A group of 6 nurses and 6 junior physician volunteers with no previous ultrasound experience underwent Graf's standard training in hands-on practice and didactic instruction. The mean age of the junior physicians and nurses was 25±1.4 and 26.3±0.8 years, respectively. All were female. The average score of the post-training test was not significantly different between the two groups.

A total of 201 newborns (402 hips or 804 hip sonograms) were examined in the study, with a male/female ratio of 118/83 and a mean age of 1.3±0.9 days at examination. Junior physicians

**Table 2. Newborn characteristics.**

| Characteristics | All<br>N = 201<br>n(%) | Junior physician- performed N = 100<br>n(%) | Nurse-performed<br>N = 101<br>n(%) | P value |
|---|---|---|---|---|
| Mean age at hip ultrasound examination, days (mean±sd) | 1.3±0.9 | 1.3±0.8 | 1.2±0.9 | 0.2877* |
| **Sex** | | | | 0.711** |
| Male | 118 (58.7) | 60 (60.0) | 58 (57.4) | |
| Female | 83 (41.3) | 40 (40.0) | 43 (42.6) | |
| Hospital | | | | 0.078** |
| Referral | 98 (48.8) | 55 (55.0) | 43 (42.6) | |
| Provincial and sub-provincial | 103 (51.2) | 45 (45.0) | 58 (57.4) | |
| Mean birth weight, kg (mean±sd) | 3.2±0.2 | 3.2±0.2 | 3.2±0.2 | 0.820* |

*t-test

**Chi$^2$ test

examined 100 newborns (200 hips or 400 hip sonograms), while nurses examined 101 newborns (202 hips or 404 hip sonograms). The study subjects/newborns were well balanced for the distribution of baseline characteristics, such as mean age, sex, birth weight, and hospitals. (**Table 2**)

As shown in **Fig 2**, of 100 newborns examined in the junior physician-performed group and 101 newborns examined in the nurse-performed group, the proportion of sufficient

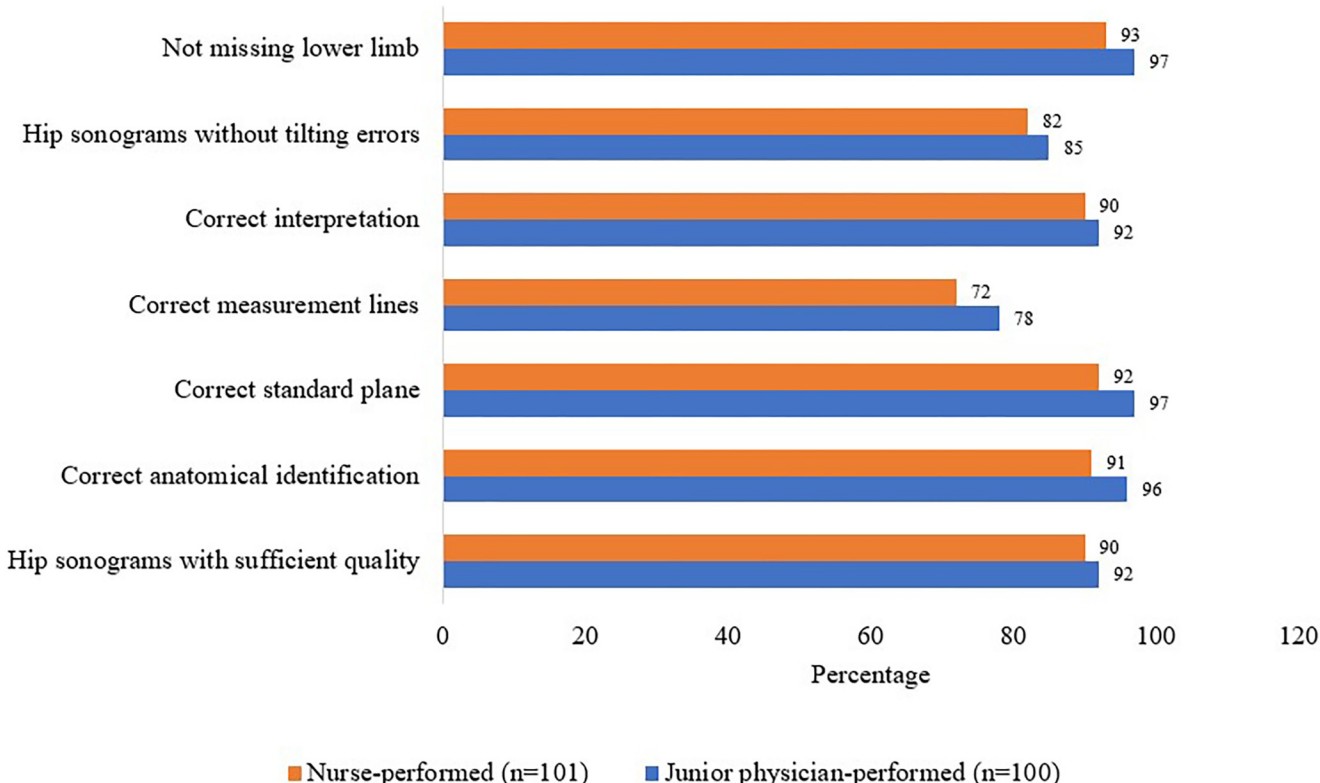

**Fig 2. Hip ultrasound examination performance of the nurses's and junior physicians' groups.**

**Table 3.  Univariate analysis of demographic and hip ultrasound examination characteristics of correct and incorrect diagnosis.**

| Characteristics | Correct diagnosis N = 182 | Incorrect diagnosis N = 19 | p-value |
|---|---|---|---|
| Age of newborns (CI) | 1.3 ± 0.9 (1.2–1.5) | 1.3 ± 0.8 (0.7–1.5) | 0.989 |
| Mean birth weight (CI) | 3.2 ±0.2 (3.2–3.3) | 3.1 ± 0.2 (2.9–3.2) | 0.012 |
| Hospital | | | 0.275 |
| UB | 91 (50.0) | 7 (36.8) | |
| rural | 91(50.0) | 12 (63.2) | |
| Performed by | | | 0.484 |
| junior physician | 92 (50.6) | 8 (42.1) | |
| nurse | 90 (49.4) | 11 (57.9) | |
| Anatomical identification | | | <0.0001 |
| correct | 182 (100.0) | 5 (26.3) | |
| incorrect | 0 | 14 (73.7) | |
| Standard plane cut | | | <0.0001 |
| correct | 182 (100.0) | 7 (36.8) | |
| incorrect | 0 | 12 (63.2) | |
| Measurement lines | | | <0.0001 |
| correct | 148 (81.3) | 2 (10.5) | |
| incorrect | 34 (18.7) | 17 (89.5) | |
| Tilting errors | | | 0.002 |
| yes | 26 (14.3) | 8 (42.1) | |
| no | 156 (85.7) | 11 (57.9) | |
| Lower limb missing | | | <0.0001 |
| yes | 0 | 11 (57.9) | |
| no | 182 | 8 (42.1) | |

*Fisher exact

quality of hip sonograms was 92.0% and 89.1% (p = 0.484), respectively. In the anatomical identification, 96 (96.0%) and 91 (90.1%) newborns' hip sonograms in the respective groups were correct (p = 0.100). The incorrect standard plane was recorded in comparatively few proportions in both groups (3.0% in the junior physician's group vs. 8.9% in the nurse's group; p = 0.077).

The proportion of measurement lines was 78.0% in the junior physician-performed group, while in the nurse-performed group, the proportion was 71.3% (p = 0.274).

Eventually, 92.0% (90) of the junior group and 89.1% (90) of the nurse group (p = 0.484) correctly diagnosed the "Group A" (Graf's Type 1 hip) or "Non-Group A" (Graf's non-Type 1 hip) (**Fig 2**).

A common error among the groups was the tilting of the transducer, which deflects the ultrasound beam and can produce a misleading and incorrect image (17.0% in the junior group vs. 16.8% in the nurse's group, p = 0.975), followed by the missing of the lower limb (3.0% in the junior group vs. 7.9% in the nurse's group, p = 0.125) (**Fig 2**).

Univariate comparisons of demographic and hip ultrasound (Graf method) examination characteristics of "correct" and "incorrect" diagnosis are displayed in **Table 3**. The incorrect diagnosis was significantly more likely to be associated with smaller birth weight of newborns (p = 0.012), incorrect anatomical identification (p<0.0001), incorrect measurement lines (p<0.0001), missing lower limb and tilting errors (p<0.002).

## Discussion

The study provides evidence that nurses could be a practical alternative where suitable hip ultrasound-trained doctors are unavailable. The "Graf's standard image" quality success high proportion of 92.0% and 89.1% were observed for junior physicians' and nurses' groups, respectively. Furthermore, the study revealed similar "Group A" or "Non-Group A" hips' correct identification rates in both groups. The most common error among the groups were tilting of the transducer and missing of the lower limb.

In developing countries, implementing and sustaining any newborn screening, including newborn hip ultrasound, is challenging due to potential political instability, lack of trained human resources and less developed public health systems, etc [21]. Over the past decade, the humanitarian Swiss-Mongolian Pediatric Project—SMOPP, commissioned by the MOH in Mongolia, has established a newborn ultrasound screening and early conservative treatment program for developmental dysplasia of the hip [4, 5]. The obligation to implement a DDH ultrasound screening program, especially in a country with limited resources, requires plausible, simple structures and straightforward algorithms. These requirements led to two paradigms in the implementation of the SMOPP: a) application of the Graf's ultrasound method [3, 10] as early as possible in the newborn period in a simplified form with only 4 intervention groups [18], and b) uniform, simple, cost-effective outpatient treatment of all DDH severity levels with a flexion-abduction device performed by the parents under pediatric surveillance [3, 10, 18].

In Mongolia, the Graf's method is used in a simplified modified ABCD adaptation [18], which combines the Graf types into intervention groups according to therapeutic aspects (A = normal; B = physiologically immature, worthy of control; C = DDH, in need of immediate therapy through flexion-abduction-braces; D = dislocated). Knowing the high maturation potential in the first 3 months of life [22], the strategy must be implemented as early as possible after birth. The feasibility of the concept in Mongolia was proven by a cohort study [4] and subsequent statistical analyses [5]. All cases (n = 107) of DDH discovered in newborns were cured using the Tübingen braces [23]. In addition, the hip ultrasound screening has been well accepted by local newborn hip ultrasound screeners (e.g., neonatologists and pediatricians) even though the screening was comparatively new to the providers. Nevertheless, the screening implementation has not been smooth, especially in prominent sub-provincial hospitals because physicians are not always available when they are on remote emergency calls from remote rural areas.

The need for a possible alternative approach to screen every newborn baby for DDH by ultrasound is obvious when trained doctors are not available, especially in rural areas. To address the challenge of detecting "non-Group A" (Graf's non-Type 1 hip) cases before discharge from maternity hospital (usually, after birth, newborns stay 1–2 days), it might be wise to train nurses since nurses are a more stable group in terms of higher numbers, mobility or changing workplace in the country [24]. This is especially crucial in rural hospitals where nomadic residents receive maternity and health services. Thus our study aimed to evaluate the feasibility of nurses without prior experience in ultrasonography to obtain the standard plane images of newborn hips. The study observed no statistically significant difference in the high quality of Graf's standard plane images between the providers: junior physicians and nurses. Nevertheless, the study suggests that the nurses might need more training to correctly identify anatomical structures.

Although the role of nurse-led hip ultrasonography is insufficiently studied, Professor Graf, the founder of the hip ultrasound method, repeatedly mentioned that the nurse-performed hip ultrasound has good accuracy. There is a study that aimed to evaluate healthcare workers'

ability to classify ultrasound images into a Graf system [25]. The study team consisted of 3 physicians and 4 nurses at the infant health care system, with no previous ultrasound experience in the Netherlands. After theoretical and practical training, seven nurses and physicians of the participating infant health centers reported their findings on hip sonograms of 80 children. This was repeated five months later. From the two evaluation moments, the intra-observer agreement and the inter-observer agreement were determined. Based on the study results, the authors concluded that the inter- and intra-observer agreement is comparable to similar studies in which the participants had a professional background in ultrasound examination. The level of agreement of the trainees in the perspective of the screening process was considered sufficient [25]. However, this study addressed only the intra- and inter-observer variability in reading a sonogram. Therefore, the findings of this study and our results cannot be compared.

On the other hand, nurses are key players in the newborn screening program, and their role is crucial in increasing newborn screening coverage [26]. Considered front liners, nurses are usually the first contact with parents in a primary care setting or health facility, allowing them to advocate and educate parents on newborn screening. Besides these advantages recently, several studies have shown that appropriately trained nurses can perform some ultrasound scans [27–32]. Although the role of nurse-led hip ultrasonography is poorly studied, it has been described as helpful in the placement of central [27] and peripheral intravenous lines [27, 29] in patients with difficult access, for the focused assessment in emergency situations [31, 33, 34] and patients with urologic [30, 35], obstetric [32, 36], and cardiac diseases [28]. In some countries, such as the United States, ultrasounds are generally performed by sonographers (technologists) and interpreted by physicians.

In the present study, common errors were missing the lower limb in both groups and subsequent measurement bias that led to the wrong diagnosis. It is mandatory to check the usefulness of a hip sonogram according to the Graf method: a) to identify the lower limb of the os ilium, b) the precise middle plane of the acetabulum roof, and c) the labrum (lower limb–plane—labrum); and errors of tilting the ultrasound transducer must be excluded. Such errors may eventually lead to incorrect procedures that may cause unnecessary over or under-treatment for babies without a quality control system [37, 38]. However, in the study, a quality control system allowed reliable review of the diagnosis and treatment of DDH and continuous education of nurses and junior doctors.

The current study has some potential limitations. 1) The generalizability of this study into the whole nurses' community of different levels of care might be limited because only six nurses and six physicians performed the examinations. However, we tried to have a delegation from different levels of care hospitals, e.g., from referral, provincial and sub-provincial hospitals. Concerning the study's limitations, the nurses' or junior physicians' characteristics might result in variations of skills related to image acquisition and interpretation which could affect the quality of the hip ultrasound scan. 2) The nurses' and junior physicians' evaluations of newborns' hips were not validated against a reference method such as computerized tomography or magnetic resonance imaging. However, Graf's method of hip ultrasound is standardized, it accepts only a standard plane cut of the acetabulum, and hence it is also reproducible. Our study focused on evaluating whether nurses and junior physicians can obtain the correct plane of the hip sonograms and identify "non-Group A" or "Graf's non-Type 1" hips. The study did not address the intra- and inter-observer agreement regarding hip sonograms. Nevertheless, the quality and interpretation of the nurse- and junior physician-performed hip ultrasound images were compared. 3) Our detection power is limited based on our power analysis (need ~15% difference to be detected).

## Conclusion

Our study provides evidence that while there might be a trend of slightly more technical mistakes in the nurse group, the overall diagnostic accuracy is similar to junior physicians after receiving standard training in Graf's hip ultrasound method. However, after a basic standard training, regular quality control is a must and all participants should receive refresher trainings. More specifically, nurses need training in the identification of anatomical structures. Despite its limitations, this study seems to confirm that nurses could be authorized to carry out hip ultrasound procedures in order to refer those newborns requiring more specialized treatment to a higher-level health establishment, and potentially be able to reduce the occurrence and prevalence of hip dysplasia and disability in developing countries.

## Supporting information

**S1 Dataset. Comparison of quality and interpretation of newborn ultrasound screening examinations for developmental dysplasia of the hip by nurses and junior physicians.** (DTA)

## Acknowledgments

We would like to thank all families, nurses and physicians who participated in this study. We would also like to thank our colleagues who provided expert advice and are members of the Swiss Mongolian Pediatric Project and the Mongolian Society of Developmental Dysplasia of the Hip Prevention.

## Author Contributions

**Conceptualization:** Munkhtulga Ulziibat, Bayalag Munkhuu, Raoul Schmid, Corinne Wyder, Thomas Baumann, Stefan Essig.

**Data curation:** Munkhtulga Ulziibat, Bayalag Munkhuu, Raoul Schmid, Corinne Wyder, Thomas Baumann, Stefan Essig.

**Formal analysis:** Munkhtulga Ulziibat, Bayalag Munkhuu, Raoul Schmid, Corinne Wyder, Thomas Baumann, Stefan Essig.

**Funding acquisition:** Munkhtulga Ulziibat, Bayalag Munkhuu, Raoul Schmid, Corinne Wyder, Thomas Baumann, Stefan Essig.

**Investigation:** Munkhtulga Ulziibat, Bayalag Munkhuu, Raoul Schmid, Corinne Wyder, Thomas Baumann, Stefan Essig.

**Methodology:** Munkhtulga Ulziibat, Bayalag Munkhuu, Raoul Schmid, Corinne Wyder, Thomas Baumann, Stefan Essig.

**Project administration:** Munkhtulga Ulziibat, Bayalag Munkhuu, Raoul Schmid, Corinne Wyder, Thomas Baumann, Stefan Essig.

**Resources:** Munkhtulga Ulziibat, Bayalag Munkhuu, Raoul Schmid, Corinne Wyder, Thomas Baumann, Stefan Essig.

**Software:** Munkhtulga Ulziibat, Bayalag Munkhuu, Raoul Schmid, Corinne Wyder, Thomas Baumann, Stefan Essig.

**Supervision:** Munkhtulga Ulziibat, Bayalag Munkhuu, Raoul Schmid, Corinne Wyder, Thomas Baumann, Stefan Essig.

**Validation:** Munkhtulga Ulziibat, Bayalag Munkhuu, Raoul Schmid, Corinne Wyder, Thomas Baumann, Stefan Essig.

**Visualization:** Munkhtulga Ulziibat, Bayalag Munkhuu, Raoul Schmid, Corinne Wyder, Thomas Baumann, Stefan Essig.

**Writing – original draft:** Munkhtulga Ulziibat, Bayalag Munkhuu, Raoul Schmid, Corinne Wyder, Thomas Baumann, Stefan Essig.

**Writing – review & editing:** Munkhtulga Ulziibat, Bayalag Munkhuu, Raoul Schmid, Corinne Wyder, Thomas Baumann, Stefan Essig.

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
