## [Decision Letter · Decision Letter 0]

24 Jul 2023

PONE-D-23-02980Comparison of quality and interpretation of hip ultrasound screening examinations by basically trained nurses to junior physicians with no previous ultrasound experiencePLOS ONE

Dear Dr. Ulziibat,

Thank you for submitting your manuscript to PLOS ONE. After careful consideration, we feel that it has merit but does not fully meet PLOS ONE’s publication criteria as it currently stands. Therefore, we invite you to submit a revised version of the manuscript that addresses the points raised during the review process.

The manuscript has been evaluated by five reviewers, and their comments are available below. The reviewers have raised a number of concerns throughout your manuscript that need attention. They recommend additional background information in the Introduction to better introduce the topic. They also feel the Methods section may benefit from further clarification, particularly as it pertains to the education of the trainees, and the inclusion of statistical power calculations which may validate conclusions drawn from your study. It is also suggested to include additional images which may enhance presentation for the reader. Could you please revise the manuscript to carefully address the concerns raised?

We look forward to receiving your revised manuscript.

Kind regards,

Richard Ali, PhD

Staff Editor

PLOS ONE

Journal Requirements:

Additional Editor Comments:

The manuscript has important value for dissemination, given the needs at the primary care level in many countries. The study and results are well designed and explained. However, perhaps from the language and writing parts there are several unclear expressions that may confuse the readers. Thus, I have made several suggestions so that you and authors can review, revise, correct and then re-submit. Thank you.

Reviewers' comments:

Reviewer's Responses to Questions

**Comments to the Author**

1. Is the manuscript technically sound, and do the data support the conclusions?

Reviewer #1: Yes

Reviewer #2: Partly

Reviewer #3: Yes

Reviewer #4: Partly

Reviewer #5: No

2. Has the statistical analysis been performed appropriately and rigorously? 

Reviewer #1: Yes

Reviewer #2: Yes

Reviewer #3: Yes

Reviewer #4: No

Reviewer #5: No

3. Have the authors made all data underlying the findings in their manuscript fully available?

Reviewer #1: Yes

Reviewer #2: Yes

Reviewer #3: Yes

Reviewer #4: Yes

Reviewer #5: Yes

4. Is the manuscript presented in an intelligible fashion and written in standard English?

Reviewer #1: No

Reviewer #2: Yes

Reviewer #3: Yes

Reviewer #4: Yes

Reviewer #5: No

5. Review Comments to the Author

Reviewer #1: There are several issues in this manuscript. Please see the attached PDF sent with the review, where I place several suggestions to improve the data analysis and manuscript writing, for better understanding and integrity.

Reviewer #2: The study designed is to provide an alternative to trained doctors when are not available in detecting DDH in newborns. It needs more clarification and efficient interpretation. The data depicted is not convincing enough. The title should be modified to more clear aim and should specify the 'newborns'. The introduction lacks the critical relevance of DDH, its diagnosis, factors, epidemiological reasons and basic information. The information regarding 'ABCD' system should be mentioned in material sand methods and well defined to the new readers. Overall, the aim of the authors to give untrained nurses to diagnose newly Borns with DDH seems a bit risky and not a reliable approach.

Reviewer #3: Dear Authors

I would like to express my appreciation for the thoroughness and quality of your work. Your research is both significant and compelling. I was impressed by the clarity of your methodology, the validity of your findings, and the overall organization of your paper.

With my acknowledgement to the strength of your results, I think that your research should include relevant figures, charts, or graphs to visually present your findings and enhance the overall presentation of the results section.

Reviewer #4: Overall promising paper. Work can be strengthened by clarification in technique/methods (more in depth description of the doctors/nurses training, handling of hips vs patients), better presentation organization, and better statistical consideration. In particular, the numbers shown suggest there may be some decrease in quality of the exam in nurses compared to doctors that might reach significance if n is increased (see below in conclusion). Power analysis is definitely needed any time a "no difference" in result is proposed.

Title:

-Might want to include what is being screened for (developmental hip dysplasia).

Article information:

-The funding information is incomplete. Need which author received the fund, and/or grant number.

Abstract:

-Base on the data, would not say “can obtain… sonograms… with equal quality.” See below in conclusion.

Introduction:

-Line 75: “drainage” of knowledge ?

Methods:

-What is the background of training of the nurses (what kind of schooling)? And of the junior physicians? What kind of physicians are they? Years of experience?

-The nurses and junior doctors completed 1 week of introductory courses and 1 week of supervised ultrasound screening. How many hours of courses were there? How many patients (approximately) were screened while supervised? You said 10-20 were “intended,” but how many were actually done?

-Was power analysis done?

-Were the right and hips treated as separate subjects? 201 newborns were screened… were the data treated as 201 subjects or 402 hips in the calculations?

-Would remove the multivariate modeling. The information in Table 4 (univariate analysis) is sufficiently clear to show the patients prone to have errors and likely source of error. The multivariate analysis does not add much, and doesn’t make sense to combined demographic and exam characteristic factors.

Results:

-The division of Table 2 and Table 3 do not make sense. Table 2 (quality of images) included diagnosis (not a quality metric) and anatomical identification (interpretative metric), but not the factors in Table 3 (tilting and missing lower limb- factors of image quality).

Conclusion:

-The numbers presented 96% correct anatomical identification for doctors vs 90% for nurses, 97% correct standard plane vs 91% for nurses nearly reach statistical significance (p = 0.077-0.10). Without power analyses, the conclusion that they are equal in quality is not justified. Would consider possibly rephrasing to say that while there might be a trend of slightly more incorrect technical factors in the nurses group, the overall diagnosis accuracy is similar.

Reviewer #5: This is an interesting topic. I have questions regarding methods.

1. Line 81 states basically trained nurses and trained junior physicians. Are they all graduated from certified medical schools? Can they represent the current nurse or junior physician population in the country?

2. Line 105 states the standard week courses. Could you specify how many hours in total? During the course, were a pretraining test and a post-training test performed? If did, how about performance?

3. Line 110 to 113 states around 10 to 20 sonograms. Is there any correlation between the training sonogram number and the final performance of each nurse and physician in the study?

4. Table 1. Why did you combine provincial and subprovincial groups? Since your paper tried to convince us that standard training could help rural regions. The subprovincial group performance is more important and better to make a point. Could you explain why you combined two groups?

5. Table 2. There are two numbers 0.245 and 0.100, on the line of Anatomical identification. Could you please explain it?

6. PLOS authors have the option to publish the peer review history of their article (what does this mean?). If published, this will include your full peer review and any attached files.

Reviewer #1: No

Reviewer #2: No

Reviewer #3: **Yes: **Mysara Rumman

Reviewer #4: **Yes: **Anderson H. Kuo

Reviewer #5: No

&nbsp

---

## [Author Response · Author response to Decision Letter 0]

21 Sep 2023

Editor Comments

The manuscript has been evaluated by five reviewers, and their comments are available below. The reviewers have raised a number of concerns throughout your manuscript that need attention. They recommend additional background information in the Introduction to better introduce the topic. They also feel the Methods section may benefit from further clarification, particularly as it pertains to the education of the trainees, and the inclusion of statistical power calculations which may validate conclusions drawn from your study. It is also suggested to include additional images which may enhance presentation for the reader. Could you please revise the manuscript to carefully address the concerns raised?

Response: Thank you and the reviewers very much for the opportunity to submit a revision of our manuscript. We added additional background information, information on the education of the trainees and statistical power calculations, as well as images. Below, we respond to each point raised by the reviewers. If we can provide any additional information or make any other changes, please do not hesitate to let us know.

The manuscript has important value for dissemination, given the needs at the primary care level in many countries. The study and results are well designed and explained. However, perhaps from the language and writing parts there are several unclear expressions that may confuse the readers. Thus, I have made several suggestions so that you and authors can review, revise, correct and then re-submit. Thank you.

Response: Thank you for your support. We are not sure which suggestions came from you as the comments of you and the reviewers were submitted in three parts (text in the e-mail, PDF and Word file). We respond to each point from all three parts below and carefully address the concerns raised.

---

## [Decision Letter · Decision Letter 1]

20 Feb 2024

PONE-D-23-02980R1Comparison of quality and interpretation of hip ultrasound screening examinations by basically trained nurses to junior physicians with no previous ultrasound experiencePLOS ONE

Dear Dr. Ulziibat,

Thank you for submitting your manuscript to PLOS ONE. After careful consideration, we feel that it has merit but does not fully meet PLOS ONE’s publication criteria as it currently stands. Therefore, we invite you to submit a revised version of the manuscript that addresses the points raised during the review process.

 Please submit your revised manuscript by Apr 05 2024 11:59PM. If you will need more time than this to complete your revisions, please reply to this message or contact the journal office at plosone@plos.org. Please include the following items when submitting your revised manuscript:A rebuttal letter that responds to each point raised by the academic editor and reviewer(s). You should upload this letter as a separate file labeled 'Response to Reviewers'.A marked-up copy of your manuscript that highlights changes made to the original version. You should upload this as a separate file labeled 'Revised Manuscript with Track Changes'.An unmarked version of your revised paper without tracked changes. You should upload this as a separate file labeled 'Manuscript'.If applicable, we recommend that you deposit your laboratory protocols in protocols.io to enhance the reproducibility of your results. Protocols.io assigns your protocol its own identifier (DOI) so that it can be cited independently in the future. For instructions see: https://journals.plos.org/plosone/s/submission-guidelines#loc-laboratory-protocols. Additionally, PLOS ONE offers an option for publishing peer-reviewed Lab Protocol articles, which describe protocols hosted on protocols.io. Read more information on sharing protocols at https://plos.org/protocols?utm_medium=editorial-email&utm_source=authorletters&utm_campaign=protocols.

We look forward to receiving your revised manuscript.

Kind regards,

Marianne Clemence

Staff Editor

PLOS ONE

Journal Requirements:

**Additional Editor Comments:**

Thank you for revising your manuscript according to the reviewers' concerns. The reviewers have a few additional minor requests to improve the quality of the language and clarity of the explanations outline below and attached. I note that you modified your title in the manuscript file - please ensure you have also updated the title in the online submission form.

Reviewers' comments:

Reviewer's Responses to Questions

**Comments to the Author**

1. If the authors have adequately addressed your comments raised in a previous round of review and you feel that this manuscript is now acceptable for publication, you may indicate that here to bypass the “Comments to the Author” section, enter your conflict of interest statement in the “Confidential to Editor” section, and submit your "Accept" recommendation.

Reviewer #1: All comments have been addressed

Reviewer #3: All comments have been addressed

Reviewer #4: (No Response)

Reviewer #5: All comments have been addressed

2. Is the manuscript technically sound, and do the data support the conclusions?

Reviewer #1: Yes

Reviewer #3: Yes

Reviewer #4: Yes

Reviewer #5: Yes

3. Has the statistical analysis been performed appropriately and rigorously? 

Reviewer #1: Yes

Reviewer #3: Yes

Reviewer #4: Yes

Reviewer #5: Yes

4. Have the authors made all data underlying the findings in their manuscript fully available?

Reviewer #1: Yes

Reviewer #3: Yes

Reviewer #4: Yes

Reviewer #5: Yes

5. Is the manuscript presented in an intelligible fashion and written in standard English?

Reviewer #1: No

Reviewer #3: Yes

Reviewer #4: Yes

Reviewer #5: Yes

6. Review Comments to the Author

Reviewer #1: This version of the manuscript contains more accurate and explanatory statements that help the reader make good sense of the important study and results. However, there are still a few very small but key additions/explanations required, again for the benefit of the reader and so there is no confusion over methods and results. Please review my accompanying PDF file, where I suggest some of these edits, and make sure somebody with good English language skill makes a final review of the manuscript, in order for it to be published.

Reviewer #3: Through my reading this manuscript, I found a clear, correct, and unambiguous language. Moreover, in this review, all the previous grammatical errors were also corrected.

Therefore, I accept this paper for publication.

Reviewer #4: Thank you for revising your manuscript and clarifying the previously noted points. The manuscript is improved and more clear. A few remaining points worth considering.

INTRODUCTION

Line 64: standard PLANE instead of plain? Might want to rephrase as "The method uses a coronal ultrasound image through the center of the acetabulum..."

Line 88, 91 Instead of "profound knowledge," would consider "good knowledge," "experienced in," or "proficient in"

METHODS

Line 119 "Other criteria ... " Would remove this sentence. It is confusing and does not add additional information.

Line 219 Would combine (1) and (2) into "difference in quality of ultrasound of 15%.

RESULT

Line 249 "the incorrect diagnosis was significantly more likely to be 'ASSOCIATED WITH' smaller birth weight of newborns..." Correlation and causation are not the same.

DISCUSSION

Would acknowledge that your detection power is limited based on your power analysis (need ~15% difference to be detected).

It might be worth noting that in some places in the world, such as the US, ultrasounds are generally performed by sonographers (technologists) and interpreted by physicians.

TABLE

Table 3: Incorrect diagnosis total should be 19, not 9

FIGURE

Figure 1A and 1B: Image in 1A is too zoomed in, making it difficult to see the position of the femoral head and worsening image quality.

Reviewer #5: Could you please correct the contradictory between line 60 and line 37? Ultrasound is an examiner-dependent modality. That is why examiners should be very trained and certified.

7. PLOS authors have the option to publish the peer review history of their article (what does this mean?). If published, this will include your full peer review and any attached files.

Reviewer #1: **Yes: **Dr. Alfredo L. Fort

Reviewer #3: **Yes: **Mysara Rumman

Reviewer #4: **Yes: **Anderson H. Kuo

Reviewer #5: No

---

## [Author Response · Author response to Decision Letter 1]

23 Feb 2024

Journal Requirements

Response: The reference list was re-checked and corrected: two mistakenly retracted papers were removed and replaced them with relevant current references (Ref 15- line: 413-417 and Ref 24: lines: 446-447), one paper was added due to additional explanations in the text (Ref 20- lines: 431-434). 

Additional Editor Comments

Thank you for revising your manuscript according to the reviewers' concerns. The reviewers have a few additional minor requests to improve the quality of the language and clarity of the explanations outline below and attached. I note that you modified your title in the manuscript file - please ensure you have also updated the title in the online submission form.

Response: Thank you very much for the opportunity to submit a revision of our manuscript. We now updated the title in the online submission form. We will respond to the points raised by the reviewers below.

---

## [Decision Letter · Decision Letter 2]

5 Mar 2024

Comparison of quality and interpretation of newborn ultrasound screening examinations for developmental dysplasia of the hip by basically trained nurses and junior physicians with no previous ultrasound experience

PONE-D-23-02980R2

Dear Dr. Munkhtulga U{lziibat,

We’re pleased to inform you that your manuscript has been judged scientifically suitable for publication and will be formally accepted for publication once it meets all outstanding technical requirements.

Kind regards,

Malgorzata Wojcik, Ph.D

Academic Editor

PLOS ONE

Additional Editor Comments (optional):

Dear Authors,

Congratulations, you have received acceptance after major and minor revision.

best wishes

Małgorzata Wójcik

Reviewers' comments:

Reviewer's Responses to Questions

**Comments to the Author**

1. If the authors have adequately addressed your comments raised in a previous round of review and you feel that this manuscript is now acceptable for publication, you may indicate that here to bypass the “Comments to the Author” section, enter your conflict of interest statement in the “Confidential to Editor” section, and submit your "Accept" recommendation.

Reviewer #3: All comments have been addressed

Reviewer #4: All comments have been addressed

Reviewer #5: All comments have been addressed

2. Is the manuscript technically sound, and do the data support the conclusions?

Reviewer #3: Yes

Reviewer #4: (No Response)

Reviewer #5: Yes

3. Has the statistical analysis been performed appropriately and rigorously? 

Reviewer #3: Yes

Reviewer #4: (No Response)

Reviewer #5: Yes

4. Have the authors made all data underlying the findings in their manuscript fully available?

Reviewer #3: Yes

Reviewer #4: (No Response)

Reviewer #5: Yes

5. Is the manuscript presented in an intelligible fashion and written in standard English?

Reviewer #3: Yes

Reviewer #4: (No Response)

Reviewer #5: Yes

6. Review Comments to the Author

Reviewer #3: The manuscript is ready to publish, no further revision needed.

I have reviewed the original manuscript and then the revised version, and decided to accept it.

Reviewer #4: (No Response)

Reviewer #5: (No Response)

7. PLOS authors have the option to publish the peer review history of their article (what does this mean?). If published, this will include your full peer review and any attached files.

Reviewer #3: **Yes: **Mysara Rumman

Reviewer #4: **Yes: **Anderson H. Kuo

Reviewer #5: No

---

## [Editor Report · Acceptance letter]

8 Apr 2024

PONE-D-23-02980R2 

PLOS ONE

Dear Dr. Ulziibat, 

I'm pleased to inform you that your manuscript has been deemed suitable for publication in PLOS ONE. Congratulations! Your manuscript is now being handed over to our production team.

Kind regards, 

on behalf of

Dr. Malgorzata Wojcik 

Academic Editor

PLOS ONE